# Development of a Medium Care Unit Using an Inexperienced Respiratory Staff: Lessons Learned during the COVID-19 Pandemic

**DOI:** 10.3390/ijerph19127349

**Published:** 2022-06-15

**Authors:** Olivier Van Hove, Alexis Gillet, Jérôme Tack, Gregory Reychler, Magda Guatteri, Asuncion Ballarin, Justine Thomas, Rolando Espinoza, Frédéric Bonnier, Michelle Norrenberg, Pauline Daniel, Michel Toussaint, Dimitri Leduc, Bruno Bonnechère, Olivier Taton

**Affiliations:** 1Department of Physiotherapy and Pneumology, Erasme University Hospital, Université Libre de Bruxelles, 1070 Brussels, Belgium; alexis.gillet@erasme.ulb.ac.be (A.G.); rolando.espinoza.laimito@erasme.ulb.ac.be (R.E.); 2Intensive Care Unit, Erasme University Hospital, Université Libre de Bruxelles, 1070 Brussels, Belgium; jerome.tack@erasme.ulb.ac.be; 3Health Economics, Hospital Management and Nursing Research Dept, School of Public Health, Université Libre de Brussels, 1070 Brussels, Belgium; 4Departement of Pneumology, Saint-Luc University Hospital, Université Catholique de Louvain, 1200 Brussels, Belgium; gregory.reychler@saintluc.uclouvain.be; 5Psychology Department, Erasme University Hospital, Université Libre de Bruxelles, 1070 Brussels, Belgium; magda.guatteri@erasme.ulb.ac.be; 6Clinical Nutrition Nurse, Erasme University Hospital, Université Libre de Bruxelles, 1070 Brussels, Belgium; asuncion.ballarin@erasme.ulb.ac.be (A.B.); justine.thomas@erasme.ulb.ac.be (J.T.); 7Department of Physiotherapy and Intensive Care, Erasme University Hospital, Université Libre de Brussels, 1070 Brussels, Belgium; frederic.bonnier@erasme.ulb.ac.be (F.B.); michelle.norrenberg@erasme.ulb.ac.be (M.N.); 8Department of Physiotherapy, Faculté des Sciences de la Motricité, Université Libre de Bruxelles, 1070 Brussels, Belgium; kine.pdaniel@outlook.com; 9Centre de Référence Neuromusculaire, Department of Neurology, Erasme University Hospital, Université Libre de Bruxelles, 1070 Brussels, Belgium; michel.toussaint@erasme.ulb.ac.be; 10Department of Pneumology, Erasme University Hospital, Université Libre de Bruxelles, 1070 Brussels, Belgium; dimitri.leduc@erasme.ulb.ac.be (D.L.); olivier.taton@erasme.ulb.ac.be (O.T.); 11REVAL Rehabilitation Research Center, Faculty of Rehabilitation Sciences, Hasselt University, 3590 Diepenbeek, Belgium; bruno.bonnechere@uhasselt.be; 12Technology-Supported and Data-Driven Rehabilitation, Data Sciences Institute, Hasselt University, 3590 Diepenbeek, Belgium

**Keywords:** intermediate care unit, middle care, education, CPAP, noninvasive ventilation, rehabilitation

## Abstract

The different waves of the COVID-19 pandemic caused dramatic issues regarding the organization of care. In this context innovative solutions have to be developed in a timely manner to adapt to the organization of the care. The establishment of middle care (MC) units is a bright example of such an adaptation. A multidisciplinary MC team, including expert and non-expert respiratory health care personnel, was developed and trained to work in a COVID-19 MC unit. Important educational resources were set up to ensure rapid and effective training of the MC team, limiting the admission or delaying transfers to ICU and ensuring optimal management of palliative care. We conducted a retrospective analysis of patient data in the MC unit during the second COVID-19 wave in Belgium. The aim of this study was to demonstrate the feasibility of quickly developing an effective respiratory MC unit mixing respiratory expert and non-expert members from outside ICUs. The establishment of an MC unit during a pandemic is feasible and needed. MC units possibly relieve the pressure exerted on ICUs. A highly trained multidisciplinary team is key to ensuring the success of an MC unit during such kind of a pandemic.

## 1. Introduction

The SARS-CoV-2 (Severe acute respiratory syndrome coronavirus 2) epidemic causing coronavirus disease 2019 (COVID-19) started in Wuhan, China, and became a global pandemic on the 11th of March 2020 [1]. COVID-19 has rapidly spread all over the world despite important efforts (i.e., lockdown, quarantine, social distancing) made to try to contain it [2]. On June 8th, the total number of detected cases was more than 533 million and the total number of deaths 6.3 million [3]. Prior to worldwide mass vaccination campaigns [4], one third of hospitalized patients developed acute respiratory distress syndrome (ARDS) requiring extensive respiratory management [5].

The emergence of new infectious diseases would accelerate in the future with an origin essentially in the wild fauna [6]. Prof. Piot, the former director of the London School of Hygiene and Tropical Medicine, stated that we are entering an ‘age of pandemics’ [7], which forces us to rethink the structures of care, research and prevention. The recent COVID-19 pandemic was the best example and has been rich for learning.

The COVID-19 pandemic forced hospitals to revise their care strategies, equipment management and logistical organization. The ultimate goal of the strategy was to avoid the congestion of intensive care units (ICUs) with COVID-19 patients. Accordingly, challenges included respiratory support outside of the ICUs and related equipment, palliative care, clinical evaluation of respiratory distress, management of dyspnea and bed availability. Belgium faced a considerable challenge as it was one of the most severely affected countries during the first waves of the pandemic, with saturated hospitals and the highest death rate per capita in the world [8].

Different actions were taken to avoid the saturation of intensive care or to compensate for the lack of beds in ICUs [9]. Considering the low availability of ventilators, continuous positive airway pressure (CPAP) devices quickly became a solution toward which all eyes were turned [10]. Fortunately, the treatment of hypoxemic COVID-19 patients with CPAP is effective [11,12] and feasible in general wards [9,11] or intermediate care units [12]. In particular, middle care (MC) units seem to be the most suitable place to set up this type of treatment and to reduce the impact on ICUs associated with the provision of optimal patient monitoring. The development of MC units is possible thanks to nurse-, physician- and physiotherapist-to-patient ratios being higher than in general wards. The staff in MC units can receive ‘step-up’ patients (defined as needing increasing care) to avoid their transfer to intensive care or ‘step-down’ patients (defined as needing decreasing care), who, conversely, come from intensive care but who are stabilized or in a weaning process from ventilation and/or tracheostomy. Outside of pandemic circumstances, MC units decrease premature discharge in general wards [13], as well as mortality [14]. However, the benefits of MC units are increased in high-risk patients [14]. MC units are traditionally used to treat patients discharged from ICUs. During the COVID-19 pandemic, however, the philosophy has been different and aimed at preventing ICUs overload [15]. There has been positive feedback regarding the use or development of MC units during a pandemic, such as COVID-19, when using this strategy [15,16]. During the COVID-19 crisis, we can distinguish three levels of organization within hospitals: (1) the COVID-19 wards, which specifically receive patients with SARS-CoV-2 who do not require care that is very different from the classic wards (surveillance, monitoring, etc.), (2) the MC—as presented in this paper—which, as mentioned above, accepts patients with higher care needs (step-up and/or step-down), and finally (3) the ICU which accepts very unstable patients often requiring intubation.

Although the multi-organ manifestations of COVID-19 are now well-documented [17], the potential long-term effects of these manifestations are still unknown. People infected with COVID-19 may develop post-infection complications. Known by a variety of names, such as long COVID or long-haul COVID, and listed in the ICD-10 classification as post-COVID-19 condition since September 2020, the manifestation and impact of this occurrence are variable. Post-COVID-19 condition occurs in people with a history of probable or confirmed SARS-CoV-2 infection, typically 3 months after the onset, with symptoms that last at least 2 months and cannot be explained by an alternate diagnosis [18]. Common symptoms include, but are not limited to, fatigue, shortness of breath, and cognitive dysfunction, which have a significant impact on daily functioning. After initial recovery from an acute COVID-19 episode, symptoms may be new or continue from the initial illness. In addition, symptoms may fluctuate or recur over time [18]. It is, therefore, of the utmost importance to develop specific units to manage these patients as soon as possible in order to reduce the burden of the infection in the acute phase but also on the long term. Early intervention has indeed been shown to be effective to reduce the risk of long COVID [19]. Different rehabilitation strategies have been proposed to enhance the function and quality of life of COVID-19-infected patients in the acute phase [20]. Rehabilitation appeared to improve dyspnea, anxiety, and kinesiophobia during the acute phase. Inconsistent results were observed for pulmonary function, whereas improvements were observed in muscle strength, walking capacity, sit-to-stand performance and quality of life [20].

In this study, we report on the development of an highly multidisciplinary and innovative MC unit in Belgium, representing an in vivo experience of what should be recommended for management during future pandemics [21]. A new respiratory virus might emerge in the future and lead to new pandemics [22].

The aim of this study was to demonstrate the feasibility of quickly developing an effective respiratory MC unit mixing respiratory expert and non-expert members from outside the ICUs. This MC was established to reduce the number of patients in ICUs, to delay the transfer of patients to ICUs and to provide treatment options to patients without access to ICUs. In this paper, we also discuss the pedagogical and logistical challenges encountered when building such MC unit.

## 2. Methods

We retrospectively analyzed the data from patients admitted to the MC unit in Erasme Hospital (Brussels, Belgium) during the second COVID-19 wave between 23 October 2020 and 15 February 2021. At that time, a 12-bed MC unit was developed to avoid the congestion of ICUs with COVID-19 patients.

### 2.1. Population

Inclusion criteria for the MC unit comprised a diagnosis of SARS-CoV-2 reverse-transcriptase polymerase chain reaction (rt-PCR)-confirmed COVID-19 complicated by respiratory distress. Respiratory distress was defined as follows: fractional inspired oxygen (FiO_2_) at 50% to maintain pulse oximetry (SpO_2_) above 92% saturation; and/or a respiratory rate above 30 cycles/min; increased work of breathing, defined as using accessory muscles and intercostal retraction; dyspnea, measured by visual analog scale; and suspicion of silent hypoxemia, the absence of dyspnea with severe hypoxemia. Only ‘step-up’ patients (defined as needing increasing care compared to those offered in COVID-19 wards) using high-flow oxygen therapy or CPAP were considered for study inclusion. Conversely, ‘step-down’ patients were transferred from the ICU to the MC unit (de-escalation of care: tracheostomy weaning, high-risk patient with ICU tetra paresis, etc.). The criteria for study exclusion are reported in the Consort flow diagram (Figure 1).

The patients included in the study were divided into two groups: patients who were considered for intubation (ICU-transferable) and patients who were not considered for intubation (not ICU-transferable). The decision not to intubate was made collegially and based on age, comorbidities and bed availability in the ICU of our institution [23].

### 2.2. Outcomes

We collected the characteristics of patients, as well as the rate of intubation and death. Comorbidity was assessed on a 10-point scale, with one point per present item. Comorbidities included the presence of chronic renal failure (CRF), chronic heart failure (CHF), arterial hypertension (ATH), diabetes, obesity, cancer, neurological disorders, cognitive disorders or a chronic respiratory disorder (COPD or lung fibrosis), as well as status as a transplant patient. We collected the number of patients treated with CPAP, the type of CPAP (Boussignac, pneumatic CPAP (PCPAP)) and interface used (mask vs. helmet).

We analyzed the overall organization of the service, including staff demographics (specialty, origin, number and staff/patient ratio) and educational variables developed to train the MC team (tools, support, seminars and coaching/debriefing).

### 2.3. Statistical Analysis

We compared the characteristics of the transferable and non-transferable patients to the ICU. The normality of each continuous variable was checked using graphical methods (boxplots, histograms and QQ plots). When normally distributed, data are presented as mean and standard deviation. For count data, absolute numbers and percentages are presented. T tests and chi-squared tests were used to compare the groups. Statistical analyses were performed at an overall significance level of 0.05.

### 2.4. Ethical Approval

This retrospective study was approved by the Ethics Commitee Erasme Hospital (P2020/659).

## 3. Results

### 3.1. Clinical Evolution

Among the 104 eligible COVID-19 patients in the MC, 52 (50%) ‘step-up’ patients were included in the study.

Forty-seven (90%) of the patients were treated with CPAP. Among these patients, 21 (45%) were treated with Boussignac, and 26 (55%) were treated with CPAP. Among the 26 CPAP users, 9 patients (31%) used masks, and 17 patients (65%) used helmets.

As shown in Table 1, 18 patients improved (35%) and 34 deteriorated (65%).

Among the 18 improving patients, 2 patients (11%) joined the general COVID unit, although they were not transferable to ICUs.

Among the 34 worsening patients, 22 (65%) had access to ICUs (15 were immediately intubated), and 12 (35%) had no access to ICUs. The latter were all admitted to palliative care with little chance of survival.

Table 1 compares the characteristics of patients according to their potential transfer to ICUs. Age, rate of intubation and death were higher in patients who were not transferable to ICUs than in those who were. There was no difference regarding the sex, body mass index (BMI) or anamnesis and drug history of patients between patients transferable and non-transferable to ICUs. PCPAP was administered to the sickest patients (see statistics in Table 1).

When comparing patients transferable to ICUs (16 + 22; 73%) to those not transferable to ICUs (2 + 12; 27%), transferable patients were younger (61 ± 11 years old, range [min = 39 years old, max = 82 years old] vs. 75 ± 11 years old, range [min = 44 years old, max = 90 years old], *p* < 0.001), had fewer comorbidities (1.7 ± 1 vs. 2.5 ± 1.2, *p* < 0.05) and lower mortality (24% vs. 86%, *p* < 0.001) than non- transferable patients. There was no difference in terms of biological variables (C-reactive protein, ferritin and D-dimers). These parameters were chosen because they are the best indicator of the severity of the disease. CRP and ferritin are the markers of inflammation, while D-dimers can be used to distinguish patients who may develop severe forms of the disease and thromboembolism [24]. There was no difference either in terms of treatment (Remdesivir, Tocilizumab, Methylprednisolone, Piperacillin/Tazobactam, Meropenem or Amoxicillin). Non-transferable patients had a higher rate of chronic respiratory disorder (64% vs. 18%, *p* < 0.004), but there was no difference regarding other comorbidities (CRF in 23% of cases, CHF: 12%, ATH: 63%, diabetes: 44%, obesity: 44%, cancer: 6%, neurological disorder: 8%, cognitive disorder 12%, transplant patient: 12% and COPD or lung fibrosis: 31%).

### 3.2. Establishment of the MC Unit

A 12-bed MC unit was developed to manage COVID-19 patients. The unit was equipped with monitoring devices (pulse oximetry, electrocardiography and signal of respiratory rate) directly connected to the nurses’ office and with cameras allowing for the monitoring of patients with CPAP.

A summary of the medical team and the reorganization of the MC are presented in Figure 2 and Table 2. The ratio of nurses to patients was 1:3 ‘all around the clock’. This ratio corresponds to the legal standard in Belgium for intensive care [25]. The background of the nurses was diverse, but most lacked experience in respiratory disease. Physiotherapists were present 24 h a day in 8-h shifts. The physicians in charge of the unit were pulmonologists.

The multidisciplinary medical team, consisting of experienced and inexperienced professionals in respiratory care, was constituted as follows.

The medical staff (MD) included pulmonologists managing the unit 24 h/7 days. They worked in cooperation with intensive care and internal medicine specialists. Because the orthopedic department was closed during the study period, the majority of nurses came from this department.

A mix of experienced and inexperienced respiratory physiotherapists worked in shifts to cover 24 h/7 days in cooperation with nurses to manage CPAP and O_2_ delivery in the COVID-19 units.

Anxiety and related dyspnea [26], cognitive and affective attitudes, fear, claustrophobic feelings and reactivation of post-traumatic experiences regarding previous use of non-invasive ventilation [27,28] were supported by our psychologists. Their role was to help decrease the symptoms of anxiety and depression in COVID-19 patients, such as numbness and sleep disturbances [29,30]. In cooperation with the social workers, they were key staff members in the management of palliative care to support patients and their families.

In our MC unit, we benefited from specific nutrition management [31] supported by published recommendations [32]. When oral intake was insufficient, enteral nutrition (EN) via a nasogastric tube (NGT) was considered, provided that the O_2_ needs were below 9 L/min and/or FiO_2_ < 60% without using a CPAP. Alternatively, parenteral nutrition (PN) was started with a peripherally inserted central catheter (PICC line).

### 3.3. Training of the Clinicians

Table 3 presents the quantitative scores and tests used to determine the respiratory distress, dyspnea, respiratory support, hygiene, respiratory parameters and adequacy of the response to the alarms of the devices. Different media were used to train and educate the MC team, such as seminars, demonstrations, real-life coaching and debriefing, and we also developed video supports.

## 4. Discussion

### 4.1. General Discussion

The main result of this study is that we found the creation of an MC unit in Belgium with a majority non-respiratory specialist staff to be feasible and efficient. In our MC unit, we admitted 52 patients requiring acute management. A total of 90% of patients benefited from CPAP, of which 38% were stabilized in the MC and were able to be transferred back to the COVID unit.

Among deteriorating patients with access to the ICU, 68% (15/22) were intubated. This demonstrates the severity of their condition and the impossibility of treating them without a proper ICU structure.

Nevertheless, during the time they were treated in the MC, they did not occupy an ICU bed. This relieved the pressure on the ICU. We observed no cases of respiratory arrest nor emergency intubation in the MC unit. This demonstrates that, thanks to the uninterrupted communication between the MC and the ICU, no late intubations occurred.

Of the patients admitted to the MC unit, 14 had no access to an ICU and 12 died. The philosophy of the management was that patients should have a chance to benefit from acute care and, if necessary, from quality palliative care. This characteristic may explain the differences between this and other studies. The number of patients who avoided ICUs (42%) appears to be lower than that reported in the literature (61%) [15]. However, only the most severe patients were admitted in our MC, regardless of whether they had access to an ICU. Indeed, some patients could be stabilized with Boussignac CPAP in COVID-19 wards and are not included in this study. The patients received maximum care in the MC unit and, if necessary, palliative care in the same unit.

The implementation of an MC unit requires specific organization; recruitment of nurses, doctors and physiotherapists from other departments; and a robust educational structure. The successful implementation of an MC unit in this case required multidisciplinary co-operation.

This study suggests that the creation of an MC with specific medical staff trained without an ICU background is possible. Of note, this type of setup requires important educational and coaching support and coordination between the different healthcare specialists. Patients admitted to this MC were in respiratory distress and required careful management. The acquisition of specific knowledge in respiratory physiology, disease management, clinical evaluation, use of CPAPs, emergency procedure and hygiene procedure was possible in a relatively short time (2 weeks). No serious incident was reported during this period.

### 4.2. Organizational Description

Twelve beds were open in our new MC unit, which is in line with reports in the literature [40]. With regard to the workload for nurses, the nurse/patient ratio was slightly higher in our MC unit (1:3) as compared to the ratio observed in other structures (1:2.5) [40]. The physiotherapist/patient ratio was 1.5:12, although one physiotherapist was present in the unit 24 h a day to monitor the patients. The presence of a physiotherapist at night is usual in our hospital [41]; however, during this period, a specific physiotherapist was engaged for the COVID-19 and MC units to support the management of CPAPs, respiratory distress and other respiratory emergencies.

### 4.3. Communication and Cooperation

There was a close co-operation between the ICU, MC and COVID units. For example, the pulmonologists attended the ICU debriefing every morning to know the number of free beds and to identify patients likely to be discharged from the ICUs. The physiotherapists carried out respiratory assessments with the physicians of the COVID units and committed them to the MC team. There were constant exchanges between nurses and physiotherapists of the ICU and MC unit in order to allow for optimal management of patients in acute situations for managing CPAPs, respiratory distress and other emergency respiratory care.

### 4.4. Training

Pedagogy and training of the healthcare personal are the key factors for the success of such a project. We know that an increase in the number of ICU beds is independently associated with increased in-hospital mortality [42]. However, adequate training is likely to reduce this phenomenon. Our MC unit received COVID-19 patients with respiratory distress requiring high oxygen flows or CPAP. However, a small number of the healthcare staff working on this unit had a respiratory background. None of the nurses had worked on a pulmonology or intensive care unit before the pandemic. Regarding the physiotherapists present during the day, 25% were respiratory specialists, 37% had an intermediate profile and 38% had a respiratory background. Important pedagogical work was, therefore, conducted. Initially, we organized half days of theory on respiratory physiology, respirator functioning and response to critical alarms. Additionally, practical work was carried out to familiarize staff with the modes of oxygen therapy and non-invasive ventilation. Having a senior presence in the respiratory field around the clock allowed for permanent coaching and debriefing. The physiotherapists received extensive training in the respiratory assessment of patients. Training included the recognition of the respiratory pattern, measurement of the respiratory rate, knowledge of the work of respiratory muscles and evaluation of dyspnea. Special attention was paid to the specificities of this disease, such as silent/happy hypoxia or patient self-induced lung injury (P-SILI).

### 4.5. Palliative Care

Palliative care is a peculiarity of this pandemic for which the MC staff were previously trained. Twelve patients died in the MC unit, corresponding to all patients with palliative care needs. This type of care is a common subject of discussion in specific guidelines for the management of COVID-19 patients [43]. To improve the quality of care of such patients, team meetings were organized, and decisions were made collegially. In the context of palliative care, specific end-of-life support was not feasible, as the degradation of patients was often so fast that it was not possible to make such arrangements. Therefore, we decided to exceptionally allow/increase family visits as soon as the medical situation deteriorated. This strategy alleviated the sense of isolation and provided emotional support to the patient and their family, avoiding pathological mourning as much as possible [44]. In the MC unit, the doctors, nurses and psychologists set up a communication plan with families for future exchanges regarding medical and psychological news, which seemed to maximize family coping and reduce their anxiety [45].

### 4.6. Perpsectives

This study opens new perspectives on how to develop an MC unit in an emergency context and allows for reflective insights into preparation for future pandemics. For example, the nurses and physiotherapists trained during the second COVID-19 wave are likely to be easily mobilized for future waves. In our experience, the MC unit, consisting of a motivated multidisciplinary staff, was not directly connected to the ICUs. Such an organization constitutes a so-called silent MC unit [21].

However, this type of MC infrastructure cannot replace traditional ICUs and their specialized staff. Many of the specialties, procedures and forms of care carried out in ICUs require highly specialized training and infrastructure.

### 4.7. Current Limitations of MCs

The reassignment of caregivers can generate stress and burden among MC staff, who face a significant cognitive overload in a stressful pandemic situation. This problem must be considered when developing such structures. The development of MC units requires the temporary closure of the department where the nurses and physiotherapists usually work, which is neither possible nor desirable in the long term.

During transfers to the ICU, some patients were very dependent on their PEEP maintaining saturation above 90% with an FiO_2_ above 80%. A special protocol was created for this situation, which allowed the patient to be transported with a PEEP and an FiO_2_ of 100%. Our protocol was similar to that used for airborne patient transfers [46]. Nevertheless, there was no certainty as to the absence of the risk of aerosolization of the virus. This undoubtedly constitutes a limitation of the creation of a middle care unit far from ICUs. Proximity can limit logistical problems for tasks, such as emergency transport and intubation under suitable conditions [47].

### 4.8. Limitations of the Study

This study is essentially descriptive, with statistics reflecting only patient severity during the second wave of the COVID-19 pandemic in the MC unit. The development of such an MC unit is not possible everywhere, as it requires services to be closed to free up nurses, doctors and physiotherapists. Nevertheless, this study represents an interesting model for the implementation of an MC unit during a pandemic.

## 5. Conclusions

The creation of an MC unit to delay ICU admissions and free up beds is feasible and appears to be a suitable option to reduce the pressure on specialized units. However, this type of MC infrastructure is not intended to replace ICUs; it is merely a way to take the pressure off. This type of project requires significant educational, logistic and psychological support. In the future, the psychological impact and cognitive load on the staff involved in respiratory care should be given special attention, as well as patient satisfaction and preference.

## Figures and Tables

**Figure 1 ijerph-19-07349-f001:**
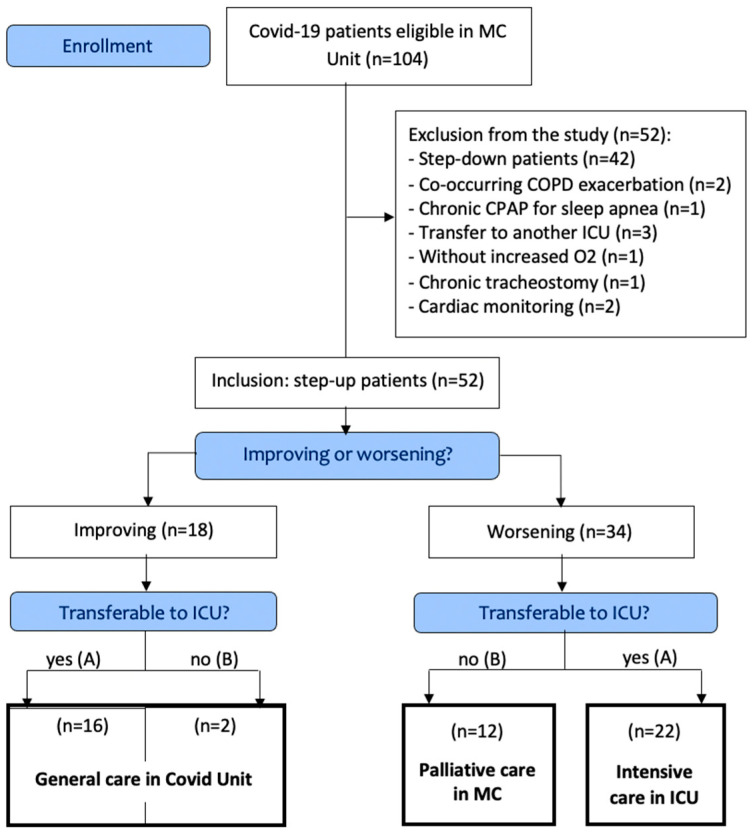
CONSORT Flow diagram of the allocation of the patients.

**Figure 2 ijerph-19-07349-f002:**
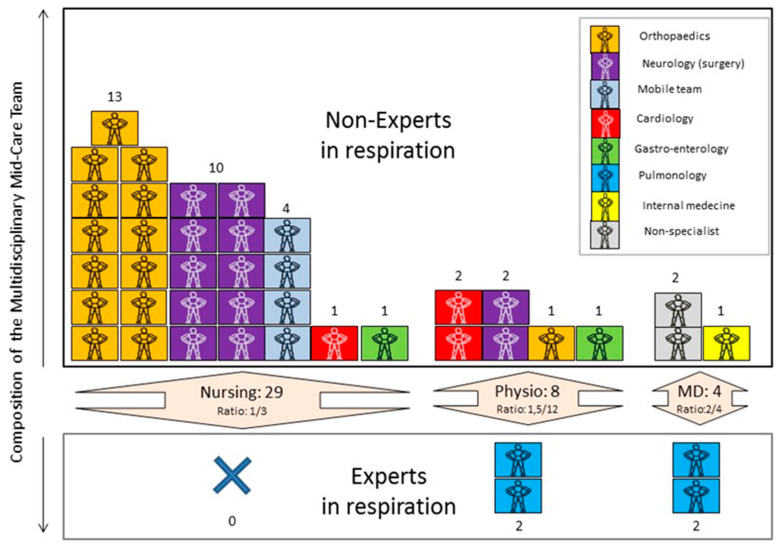
Reorganization of the Middle Care team during the COVID-19 pandemic.

**Table 1 ijerph-19-07349-t001:** Characteristics of transferable to ICU versus not transferable to ICU patients.

	All Patients	ICU Transferable (A)	ICU Non-Transferable (B)	*p*-Value
Patients, N [%]	52	38 [73%]	14 [27%]	/
Sex; female [%]	25 [48%]	31 [82%]	8 [54%]	0.148
Age (years), mean (SD)	65 (12)	61 (11)	75 (11)	<0.001
BMI (kg/m^2^), mean (SD)	27.4 (5)	35 (5)	26.9 (6)	0.655
Intubated, N [%]	15 [29%]	15 [39%]	0	/
Death, N [%]	21 [40%]	9 [24%]	12 [86%]	<0.001
COMORBIDITIES				
Comorbidities (10 points scale), mean (SD)	1.9 (1.2)	1.7 (1)	2.5 (1.2)	0.025
CRF, N [%]	12 [23%]	8 [21%]	4 [28%]	0.842
CHF, N [%]	6 [12%]	2 [5%]	4 [28%]	0.065
ATH, N [%]	33 [63%]	21 [55%]	12 [86%]	0.089
Diabetes, N [%]	23 [44%]	18 [47%]	5 [36%]	0.663
Obesity, N [%]	15 [29%]	11 [29%]	4 [28%]	1
Cancer, N [%]	3 [6%]	1 [3%]	2 [14%]	0.353
Neuro., N [%]	4 [8%]	2 [5%]	2 [14%]	0.619
Cognitive disorders, N [%]	6 [12%]	3 [8%]	3 [21%]	0.387
Transplantation, N [%]	6 [12%]	5 [13%]	1 [7%]	0.910
Respiratory disorders, N [%]	16 [31%]	7 [18%]	9 [64%]	0.004
MEDICATIONS				
Remdesivir, N [%]	7 [13%]	5 [13%]	2 [14%]	1
Tocilizumab, N [%]	6 [11%]	5 [13%]	1 [7%]	0.910
Methylprednisolone, N [%]	52 [100%]	38 [100%]	14 [100%]	1
Piperacillin/Tazobactam, N [%]	20 [38%]	14 [37%]	6 [43%]	0.941
Meronem, N [%]	4 [8%]	3 [8%]	1 [7%]	1
Amoxicillin, N [%]	12 [23%]	8 [21%]	4 [28%]	0.841

BMI: body mass index; CRF: chronic renal failure; CHF: chronic heart failure; ATH: arterial hypertension; Neuro: neurological disorders.

**Table 2 ijerph-19-07349-t002:** Composition and organization of the Middle Care team.

Profession	N	Timetable	Original Department	Caregivers/Patients Ratio
Nurse	29	24/7	Orthopedic and ORL (*n* = 13) Neurology (*n* = 10) Mobile Team (*n* = 4) Cardiology (*n* = 1) Gastroenterology (*n* = 1)	1/3
Physiotherapists	8	24/7	Pneumology (*n* = 2) Cardiology (*n* = 2) Gastroenterology (*n* = 2) Neurology (*n* = 2) Orthopedic (*n* = 1)	1.5/12
Night physiotherapist	2 per night	Night	Intensive Care Unit (*n* = 2)	1 for ICU (30 beds) 1 for MC and COVID unit
Occupational therapist	2	Day	Neurology (*n* = 2)	1/12
Medical doctor	5	24/7	Pneumology (*n* = 2) Internal medicine (*n* = 1) Postgraduate (*n* = 2)	2/4
Psychologist	2	Day	/	/
Social worker	2	Day	/	/
Cleaning and maintenance	1	Day	/	/

**Table 3 ijerph-19-07349-t003:** Educational setup used when developing the MC.

Domain	Theme	Items	Theory	Demo.	Coaching	Video
Physiology	Respiratory system and anatomy	Lung, ventilation	X			
	Theoretical basis of ventilatory supports	CPAP, invasive and non-invasive ventilation	X			
Respiratory distress	Respiratory rate	Range [33]	X	X	X	
	Use of accessory inspiratory muscles	Palpation of phasic contraction [34]	X	X	X	
	Paradoxical breathing	Visual and palpation [34]	X	X	X	
	Face examination	Fear, effort [34]	X	X	X	
	Respiratory pattern	Thoracoabdominal regularity [34]	X	X	X	
Dyspnea	Communicating patient	Borg, part of MDP (work, air hunger) [35]	X		X	
	Non-communicating patient	IC-RDOS [36]	X		X	
Respiratory support and Oxygenotherapy	HFNO	FiO_2_, flow	X	X	X	X
	CPAP	Boussignac, Sleep apnea, Drager, Nasobuccal mask, helmet	X	X	X	X
Hygiene	Aerosolizing procedure		X	X	X	X
	Aerosolized treatment [37]		X	X	X	X
Parameters	SPO_2_/FiO_2_ [38]		X			
	ROX index [39]		X			
	PAO_2_/FiO_2_		X			

Borg: Borg dyspnea score; MDP: Multidimensional Dyspnea Profile; IC-RDOS: intensive care respiratory distress observation scale; HFNO: high-flow nasal oxygen; ROX index: ratio of saturation in arterial blood and fractional inspired oxygen to respiratory rate; Demo: demonstrations; Video: video supports. X indicates materials available.

## Data Availability

The data that support the findings of this study are available on request from the corresponding author (Olivier Van Hove).

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
