# Peer review of "Development of a Medium Care Unit Using an Inexperienced Respiratory Staff: Lessons Learned during the COVID-19 Pandemic"

_ijerph, 2022, doi:10.3390/ijerph19127349_

Round 1

Reviewer 1 Report

Review of Hove et al, “Development of a Medium Care Unit using an inexperienced respiratory staff: lessons learned during the COVID-19 pandemic” in MDPI Env Res and Pub Health

The authors look at the role of middle care (MC) units for the treatment of COVID. It is a descriptive paper without a lot of investigation. However, this is an important topic in the treatment of COVID and could inform for future respiratory viral outbreaks. Due to the nature of the paper there is not a lot of suggestions for improvement as it basically summarizes what was seen. Of note, there is very little in the literature on this approach raising the interest in this publication.

Major points:

1-    COVID-19 disease/pandemic/history/outbreak needs to be included in the introduction even though it might seem like old news to provide context to the paper.

2-    A brief description of which and why the biological variables (C-reactive protein, ferritin and D-dimers) were chosen is needed.

3-    They provide age as a mean which is fine but I would like to also see the range of ages in each group.

Minor points:

1-    Prof Piot, needs an introduction/support for why he an expert. I am not sure he is able to actual predict an “age of pandemics”.

Reviewer 2 Report

The paper “Development of a Medium Care Unit using an inexperienced respiratory staff: lessons learned during the COVID-19 pandemic” shows how difficult the time was for health services during the successive waves of the pandemic. However, the paper, particularly the section on patients admitted to the MC ward, is very chaotic. The following is a list of comments:

1. first of all, it should be mentioned in the description of MC wards that they are structures created in Belgium. This information does not appear at all in the Abstract and very late in the Introduction.

2. line 60 - the term "across the country" refers to the country, because during the height of the pandemic basically every country did what it could, but the actions were not coordinated worldwide. Up to this point, it does not yet appear that the study concerns Belgium.

3. In the Introduction there should be a description of the structure of the different departments, e.g. covid - MC - ICU. Such an introduction will help the reader to understand the sequence of events, e.g. lines 169 and 238.

4. line 106 - has the research described in this manuscript already been published? This is what appears from this paragraph.

5. lines 128 and 130 - the terms "step-up" patients and "step-down" patients should be described in more detail

6. section 3.1 - the information about the patients is described extremely chaotically. Line 166 - data are given in percentages and then later in the text the description refers to numerical data. What is the point of indicating more important data as %? It is the % data that should be in brackets.

7. lines 168 - 172 - In one paragraph there is information about a group of patients who got better and who did not, each of these pieces of information is important. It is better to separate the two groups of patients and describe only the improved in one section and the deteriorated in the next. Do not mix them together.

8. line 175 - such information does not result from the table, because it is not included there.

9. lines 188-190 - who does this data refer to?

10. The text does not mention Table 2, which is a pity as it is more readable than Figure 2

11. line 251 - "appears to be lower than that reported in the literature" - what data is described in the literature?

Reviewer 3 Report

The abstract should be a total of about 200 words maximum. Yours have 246 words, please reduce the word count.

Please clarify the aim of the study in the abstract.

The introduction adequately describes the actual knowledge in the study’s matter.  The aim of the study is formulated.

line 186 please provide the international names of drugs, not drugs trade names

Round 2

Reviewer 2 Report

The Authors have responded to all my comments. I have no further objections to the work.